# Can Large Language Model Help Design Effective Neural Operators for Solving Partial Differential Equations?

## Abstract

Neural operators promise mesh and resolution independent surrogates for solving partial differential equations, yet building architectures that respect equation structure and train reliably still requires substantial expert effort. We ask whether a large language model can design neural operators end to end. We present a four agent pipeline with roles Theorist, Programmer, Critic, and Refiner. The Theorist selects a mathematically grounded operator for a user specified PDE and derives its formulation. The Programmer produces a self contained PyTorch implementation. The Critic performs adversarial review to expose numerical and software issues. The Refiner applies targeted corrections. An automated PDE solver completes the loop by generating data, training the synthesized model, and reporting evaluation metrics and plots. Across extensive PDE benchmark problems, the LLM designed operators consistently outperform strong baselines and prior SOTA in accuracy and sample efficiency, while remaining stable under varied discretizations and noisy initial conditions. Ablation studies show that the Critic and Refiner steps are essential for numerical stability and generalization. These results suggest that LLMs can act as principled collaborative designers of PDE operators, translating problem statements into executable and competitive architectures and moving toward automated and theory-aware scientific machine learning.

## 1 Introduction

Neural network (NN)–based solvers have shown great promise for efficiently solving nonlinear partial differential equations (PDEs) (Li et al., 2020a; Um et al., 2020; Xu & Darve, 2020; Song & Jiang, 2023; Smith et al., 2020). Although NN-based approaches can produce fast and accurate PDE solutions, a key limitation is that they are typically trained at a specific resolution, which leads to poor generalization to problems at other resolutions. While Bar-Sinai et al. (2019) proposed an NN-based method that learns discretizations of a given PDE from fine to coarse resolutions, it cannot be directly extended to PDEs with different forms or coefficients. Overall, most NN-based approaches must be retrained to handle various resolutions. This motivates the development of resolution-free variants of NN solvers. Noting that standard NN-based methods often rely on prior knowledge (e.g., PDE forms, discretization schemes, coefficients, and boundary/initial conditions), operator learning has been proposed to train neural operators that learn mappings from parameter/function spaces to solution spaces with minimal prior knowledge of the PDE (Lu et al., 2019; Li et al., 2020b; Tripura & Chakraborty, 2023; Gupta et al., 2021; Wang & Wang, 2024).

Among these neural operators, we observe that the top performers differ across PDE problem types. For example, Wang & Wang (2024) reported that their Latent Neural Operator (LNO) exhibits $1.08\times$ and $2.42\times$ relative $\ell^2$ error compared to transolver (Wu et al., 2024) on the airfoil and plasticity datasets (Li et al., 2023), respectively. Furthermore, another key observation is that the performance of neural operators can improve or degrade even with minor architectural changes. To date, the design of neural-operator architectures, often guided by intuition, expert experience, and trial-and-error, has been "more of an art than a science" (Sanderse et al., 2025). Although neural operators such as the Fourier Neural Operator (FNO) (Li et al., 2020b) and DeepONet (Lu et al., 2019) are grounded in theoretical insights, creating novel components that align with these insights still relies

heavily on human expertise. Therefore, theory-driven design of neural operators requires human engagement and remains far from fully automated.

Large language model (LLM) agents have shown great promise in automating processes via human interactions, including mobile tasks (Wen et al., 2024; Guan et al., 2024), hardware design (Xu et al., 2024b), scientific discovery (Zimmermann et al., 2025; Filimonov, 2024; Aamer et al., 2025), code generation (Koziolek et al., 2024; Xu et al., 2024a), hyperparameter tuning (Zhang et al., 2023), and mathematical problem solving (Bian et al., 2025). For solving PDE problems, hybrid approaches (Zhou et al., 2025; Lorsung & Farimani, 2024) incorporate LLMs as components within neural architectures to improve performance. These approaches do not involve process automation, and their architectures are not designed by LLMs. In prior literature, fully automated LLM agents for PDEs include CodePDE (Li et al., 2025) and PINNsAgent (Wuwu et al., 2025). Li et al. (2025) proposed LLM agents that generate and evaluate the code of traditional PDE solvers, whereas PINNsAgent generates code on top of the physics-informed neural network (PINN) architecture. Both methods design the architectures of traditional and PINN-based solvers primarily based on numerical performance rather than theoretical insights. To the best of our knowledge, no prior work has focused on designing neural operators end-to-end with LLM agents guided by theoretical insights.

To bridge the research gaps, we ask the following question question:

*Can LLMs design accurate and efficient neural operators driven by theoretical insights?*

In this work, we propose a four-agent pipeline with distinct roles: Theorist, Programmer, Critic, and Refiner. The Theorist selects an appropriate neural-operator architecture and derives the mathematical foundation specific to the given PDE problem; the Programmer translates the Theorist's insights into an efficient and clean PyTorch implementation. The Critic, serving as a skeptical but fair reviewer, further analyzes potential issues in both the Theorist's results and the PyTorch implementation and provides suggestions. Finally, the Refiner addresses all issues raised by the Critic and updates the PyTorch implementation accordingly. Across comprehensive PDE benchmark datasets, our main findings are:

1. In general, our proposed LLM agent can design mathematically grounded neural operators that outperform strong baselines and state-of-the-art human-designed neural operators for a variety of PDE problems on regular and irregular geometries;

2. The Critic and Refiner collaborate to improve the rigor of the theory, numerical stability, and generalization;

3. For obscure theories, our LLM agent may not be able to translate them into effective neural operators. Based on human expert evaluation, we find that LLMs often hallucinate and overfit to linguistic similarity while ignoring functional equivalence in these cases.

The key contributions of this work are: (i) framing the design of neural operators as a coupled process of theory, implementation, review, and refinement-providing the first paradigm in the literature that transforms this task from an art into a science; (ii) proposing the first automated, theory-aware LLM agent that consistently outperforms existing human-designed baselines across all benchmark datasets; and (iii) showing that the LLM agent is generally reliable across most mathematical theories, with hallucinations arising in the case of obscure theories.

## 2 THE PROPOSED LLM AGENT FRAMEWORK

### 2.1 NEURAL OPERATORS

Neural operators is a mesh- and resolution- independent neural architectures that learn the mapping from the parameter space to the solution space of the PDE problems. In general, consider a PDE defined on a spatial domain $\Omega \subset \mathbb{R}^d$ and a time interval $(0, T]$:

$$\mathcal{L}_a[u(x,t)] = f(x,t), \quad \forall(x,t) \in D \times (0,T], \tag{1}$$

which is subject to a set of initial and boundary conditions. Here, the parameter function $a \in \mathcal{A}$ specifies the coefficients and initial and boundary conditions of Equation 1. In operator learning,

our goal is to construct an accurate approximation for $\mathcal{G} : \mathcal{A} \to \mathcal{F}(D \times [0, T])$, which maps the parameter function $a$ to the corresponding solution function $u(x, t) \in \mathcal{F}$, via a parametric mapping $\mathcal{G}_\theta$. The aim is to learn $\theta$ such that $\mathcal{G}_\theta \approx \mathcal{G}$ from a set of training data $\{(a_j, u_j)\}_j$.

## 2.2 FRAMEWORK OVERVIEW

Designing neural operators from a scientific perspective requires several core steps: (i) select or propose a strong neural-operator backbone such as the FNO (Li et al., 2020b) or DeepONet (Lu et al., 2019); (ii) select or propose an appropriate mathematical theory that guarantees desirable properties (e.g., convergence, approximation error, function spaces, etc.); (iii) update the backbone architecture to align with the chosen theory; and (iv) implement and debug the code. As illustrated in Figure, our framework employs a four-agent pipeline to automate this workflow.

**System Prompt.** Prior studies show that role-playing instructions enable LLMs to collaborate under distinct roles, improving performance, particularly in code generation (Carlander et al., 2024; Dong et al., 2024; Takagi et al., 2025). Building on this idea, we assign the following roles via the system prompt: Theorist, a world-class research mathematician specializing in scientific machine learning; Programmer, an expert PyTorch developer in scientific machine learning; Critic, a skeptical but fair adversarial reviewer for a top AI conference; and Refiner, an expert PyTorch developer focused on debugging and refining complex models.

**Step 1: Problem Specification.** For a given PDE problem, we first translate the mathematical formulation equation 1 into natural language that LLMs can readily understand. This natural-language specification includes the problem name, equation, spatial domain, time interval, initial conditions, boundary conditions, and source terms. Instead of presenting this information in a paragraph, we use a concise, structured natural-language format, which is effective in our framework. For example, we represent 1-D Burgers' equation as:

---

**Problem Statement**

**PDE Specification**
---

**name**: 1D Burgers' Equation

**equation_latex**:
$$\frac{\partial u}{\partial t} + u \frac{\partial u}{\partial x} = \nu \frac{\partial^2 u}{\partial x^2}$$

**domain**: $x \in (0, 1)$, $t \in (0, 1)$

**initial_condition**: $u(x, 0) = u_0(x)$.

**boundary_conditions**: Periodic

**viscosity**: 0.01
---

---

**Step 2: Propose Mathematical Theory (Theorist).** The Theorist's ultimate task is to provide rigorous theoretical results to improve the performance of the selected backbone. First, we provide the Theorist with a factory of existing neural operators, including FNO (Li et al., 2020b), DeepONet (Lu et al., 2019), Transolver (Wu et al., 2024), and LNO (Wang & Wang, 2024), as well as classical architectures such as the variational autoencoder (Kingma et al., 2013) and the Transformer (Li et al., 2022). We also allow the Theorist to utilize other backbones of its choosing. We then prompt the Theorist to develop clear, rigorous, and efficient mathematical formulations that improve the selected backbone architecture for the specific problem. In this way, the Theorist offers a complete formulation and provides a tailored design of an updated neural architecture in natural language and

mathematical form. Finally, we prompt the Theorist to justify its choices and to implement self-correctness checks as determined by the Theorist. The output of this step is a script that derives the theoretical results and descriptions of the proposed neural operator.

**Step 3: Code Generation (Programmer).**   After the Theorist provides the detailed formulation and instructions, we prompt the Programmer to translate them into code for the proposed neural operator, along with any necessary helper functions and package dependencies. We instruct the Programmer to generate the code in PyTorch due to its wide use in the machine learning community.

**Step 4: Review Theoretical Results & Implementation (Critic).**   Motivated by the peer-review mechanism of AI conferences, we prompt the Critic to (i) critically analyze the mathematical derivation provided by the Theorist and the corresponding PyTorch code from the Programmer, and (ii) identify potential inconsistencies in the derivation, edge cases, numerical instabilities, and inefficiencies in the implementation. Finally, we instruct the Critic to provide a structured list of potential issues and concrete suggestions to improve the proposed neural operator.

**Step 5: Refine Code (Refiner).**   The Refiner updates the implementation of the proposed neural operator to address all issues and suggestions identified by the Critic.

**Step 6: Code Execution.**   After the updated Python code is executed, any bugs are recorded and reported back to the Critic, who identifies issues and provides suggestions. The Refiner then revises the code accordingly, and this process repeats until no bugs remain.

## 3 NUMERICAL EXPERIMENTS

**Datasets.**   We evaluate the performance of neural operators designed by our proposed LLM-agent framework across six benchmark datasets:

1. **Darcy Flow** (Li et al., 2020b): Represents 2D flow through porous media. The PDE is discretized on a $421 \times 421$ grid and downsampled to $85 \times 85$. Inputs are coefficient fields $a(x)$, and outputs are solutions $u(x, t)$. The dataset contains 1,000 training and 200 testing samples with varying medium structures.

2. **Navier-Stokes** (Li et al., 2020b): Models the 2D incompressible Navier–Stokes equation in vorticity form on the unit torus. Each sample is a $64 \times 64$ spatiotemporal field with 20 frames, where the first 10 frames are used to predict the next 10. The dataset consists of 1,000 training and 200 testing samples.

3. **Elasticity** (Li et al., 2023): Predicts the internal stress of an elastic material discretized into 972 points. Each input is a $972 \times 2$ tensor of point positions, and the output is a $972 \times 1$ tensor of stresses. The dataset contains 1,000 training and 200 testing samples.

4. **Plasticity** (Li et al., 2023): Focuses on predicting the deformation of a plastic material under a die of arbitrary shape. The input is a structured mesh of size $101 \times 31$, and the output is the deformation over 20 timesteps, recorded as a $20 \times 101 \times 31 \times 4$ tensor. The dataset includes 900 training and 80 testing samples.

5. **Pipe** (Li et al., 2023): Estimates horizontal fluid velocity within pipes represented as a structured mesh of size $129 \times 129$. The input tensor ($129 \times 129 \times 2$) encodes positions, while the output tensor ($129 \times 129 \times 1$) gives velocity values. The dataset has 1,000 training and 200 testing samples.

6. **Airfoil** (Li et al., 2023): Concerns transonic flow over airfoils governed by the Euler equations. Inputs are structured meshes of size $221 \times 51$, and outputs are Mach number fields. The dataset includes 1,000 training and 200 testing samples derived from various airfoil designs.

**Metrics.**   We train and evaluate the designed neural operators using the relative $\ell^2$ error:

$$\text{relative } \ell^2 \text{ error } = \frac{1}{N} \sum_{i=1}^{N} \frac{\|\mathcal{G}(a_i) - G(a_i)\|_{L^2}}{\|G(a_i)\|_{L^2}}, \tag{2}$$

where $N$ denotes the number of samples.

Furthermore, we evaluate the correctness and rigor of the mathematical formulations produced by the Theorist through human expert review. The review was conducted by three independent PhD candidates specializing in computational mathematics and neural operators (who are not authors of this paper). The rubric was a binary "Yes/No" judgment based on two criteria: 1) "Is the Theorist's mathematical formulation (e.g., the derivation) correct and sound?" and 2) "Is the connection between the chosen theory and the target PDE justified and logical?"

**Baselines.** We compare LLM-designed neural operators to $10+$ strong baselines and state-of-the-art (SOTA) models designed by humans, including FNO (Li et al., 2020b), U-FNO (Wen et al., 2022), F-FNO (Tran et al., 2021), LNO (Wang & Wang, 2024), ONO (Xiao et al., 2023), WMT (Gupta et al., 2022), Galerkin (Cao, 2021), LSM (Wu et al., 2023), OFormer (Li et al., 2022), Transolver (Wu et al., 2024), and LaMO (Tiwari et al., 2025).

**Experimental Settings.** We evaluate several LLMs in our framework, such as gpt-5, gpt-5-mini, and the reasoning models o1 and o3. All experiments are conducted on a Linux workstation running Ubuntu (kernel 6.14, glibc 2.39) with Python 3.13.5 (Anaconda), PyTorch 2.8.0+cu129 (CUDA 12.9), an AMD Ryzen 9 9950X (16-core) processor, and a single NVIDIA GeForce RTX 4090 (48 GB) GPU.

## 4 RESULTS AND ANALYSIS

### 4.1 CAN LLMS DESIGN NEURAL OPERATORS?

We evaluate the capacity of LLM-designed neural operators across six benchmark datasets and find that they outperform existing SOTA models on five of the six datasets (Table 1). Notably, the LLM generates diverse neural architectures tailored to different datasets, underscoring its adaptability across a wide range of tasks. In terms of accuracy, LLM-designed neural operators decrease the relative $\ell^2$ error by approximately $\sim 6\%$, depending on the specific problem. Moreover, they demonstrate superior efficiency, achieving a 30-50% reduction in computational time and requiring two to three orders of magnitude fewer parameters (Figure 1). These results suggest that LLMs are capable of designing neural operators that are both efficient and accurate, grounded in theoretical principles. As a side note, while LLM-generated neural operators do not achieve the highest performance on the Darcy dataset, this trade-off in accuracy is made in favor of improved efficiency (see Figure 1(a)).

Table 1: Relative $\ell^2$ error comparisons of LLM-designed neural operators with baselines across six benchmark datasets. Lower relative $\ell^2$ error is better.

| Models | Elasticity | Plasticity | Airfoil | Pipe | N-S | Darcy |
|---|---|---|---|---|---|---|
| FNO (Li et al., 2020b) | 0.0229 | 0.0074 | 0.0138 | 0.0067 | 0.0417 | 0.0052 |
| U-FNO (Wen et al., 2022) | 0.0239 | 0.0039 | 0.0269 | 0.0056 | 0.2231 | 0.0183 |
| F-FNO (Tran et al., 2021) | 0.0263 | 0.0047 | 0.0078 | 0.0070 | 0.2322 | 0.0077 |
| LNO (Wang & Wang, 2024) | 0.0052 | 0.0029 | 0.0051 | 0.0026 | 0.0845 | 0.0049 |
| ONO (Xiao et al., 2023) | 0.0118 | 0.0048 | 0.0061 | 0.0052 | 0.1195 | 0.0076 |
| WMT (Gupta et al., 2021) | 0.0359 | 0.0076 | 0.0075 | 0.0077 | 0.1541 | 0.0082 |
| Galerkin (Cao, 2021) | 0.0240 | 0.0120 | 0.0118 | 0.0098 | 0.1401 | 0.0084 |
| LSM (Wu et al., 2023) | 0.0218 | 0.0025 | 0.0059 | 0.0050 | 0.1535 | 0.0065 |
| OFormer (Li et al., 2022) | 0.0183 | 0.0017 | 0.0183 | 0.0168 | 0.1705 | 0.0124 |
| Transolver (Wu et al., 2024) | 0.0062 | 0.0013 | 0.0053 | 0.0047 | 0.0879 | 0.0059 |
| LAMO (Tiwari et al., 2025) | 0.0050 | 0.0007 | 0.0041 | 0.0038 | 0.0460 | 0.0039 |
| LLM (gpt-5) | 0.0049 | 0.0018 | 0.0043 | 0.0030 | 0.0387 | 0.0132 |
| LLM (gpt-5-mini) | 0.0051 | 0.0023 | 0.0052 | 0.0032 | 0.0420 | 0.0134 |
| LLM (o1) | 0.0047 | 0.0007 | 0.0041 | 0.0023 | 0.0389 | 0.0068 |
| LLM (o3) | 0.0049 | 0.0007 | 0.0038 | 0.0022 | 0.0512 | 0.0064 |

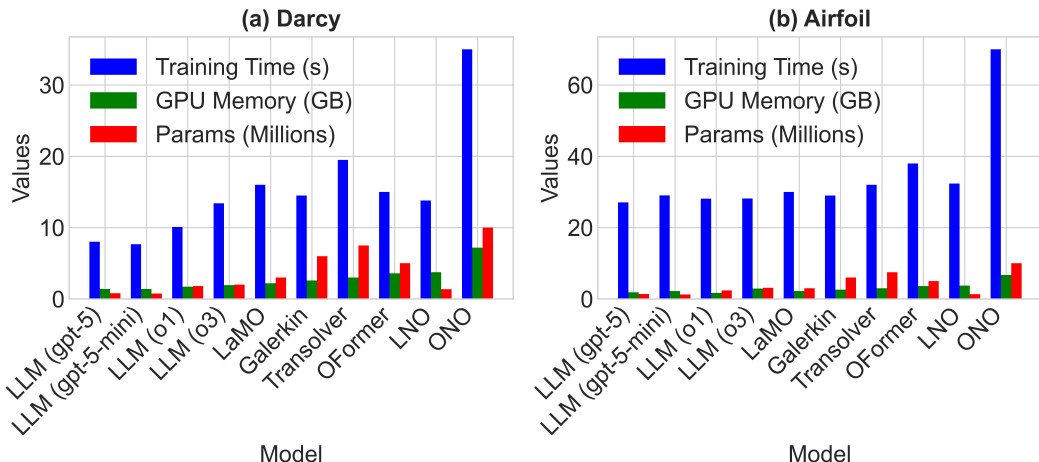

Figure 1: Comparison of computational efficiency including training time (sec per epoch), GPU memory (GB), and parameters count (M) on (a) Darcy and (b) Airfoil datasets.

## 4.2 DOES THEORY-AWARE DESIGN PROVIDE BENEFITS?

To further explore the contribution of the theoretical insights provided by the Theorist in the design process, we conduct ablation studies comparing our LLM-agent framework to a variant without the Theorist. In the latter, without theoretical guidance, the LLM agents generate neural-operator code directly (as in (Wuwu et al., 2025; Li et al., 2025)), and the Critic reviews only the numerical aspects. In this way, neural operators are designed in an art-driven rather than theory-aware paradigm. To evaluate the effectiveness of theory-aware design, we measure the performance of both frameworks in terms of accuracy and generalization.

| Model | Elasticity | Plasticity | Airfoil | Pipe | N-S | Darcy |
|---|---|---|---|---|---|---|
| **With Theorist** | | | | | | |
| LLM (gpt-5) | 0.0049 | 0.0018 | 0.0043 | 0.0030 | 0.0387 | 0.0132 |
| LLM (gpt-5-mini) | 0.0051 | 0.0023 | 0.0052 | 0.0032 | 0.0420 | 0.0134 |
| LLM (o1) | 0.0047 | 0.0007 | 0.0041 | 0.0023 | 0.0389 | 0.0068 |
| LLM (o3) | 0.0049 | 0.0007 | 0.0038 | 0.0022 | 0.0512 | 0.0064 |
| **Without Theorist** | | | | | | |
| LLM (gpt-5) | 0.0082 | 0.0047 | 0.0134 | 0.0094 | 0.1630 | 0.0188 |
| LLM (gpt-5-mini) | 0.0086 | 0.0055 | 0.0134 | 0.0116 | 0.1635 | 0.0192 |
| LLM (o1) | 0.0079 | 0.0026 | 0.0098 | 0.0102 | 0.1630 | 0.0106 |
| LLM (o3) | 0.0077 | 0.0031 | 0.0104 | 0.0103 | 0.1639 | 0.0117 |

Table 2: Relative $\ell^2$ error comparisons of neural operators designed by LLM frameworks with and without Theorist across six benchmark datasets. Lower is better.

Table 2 reports the relative $\ell^2$ errors of neural operators obtained with and without the Theorist. In general, neural operators designed by the framework with the Theorist exhibit lower errors, demonstrating the effectiveness of incorporating theoretical insights into the design process. Moreover, the improvements are particularly evident on more complex benchmark datasets, such as *Airfoil* and *Pipe*, where the error differences between the two frameworks range from approximately $3\times$ to $5\times$.

To further examine the generalization of theory-aware neural-operator design via LLMs, we follow the experimental setting in Wang & Wang (2024) and downsample the Darcy dataset from a resolution of $421 \times 421$ to $241 \times 241$, $211 \times 211$, $141 \times 141$, $85 \times 85$, $61 \times 61$, and $43 \times 43$. Neural operators are trained on the $43 \times 43$ dataset and tested on the others. Figure 2 shows that theory-aware neural operators consistently outperform those without theoretical insights across all resolutions for not only structured grids (e.g., the Navier-Stokes example) but also irregular geometries (e.g., the Elasticity, Airfoil, Pipe, and Plasticity examples). This implies that the LLM design, which incorporates

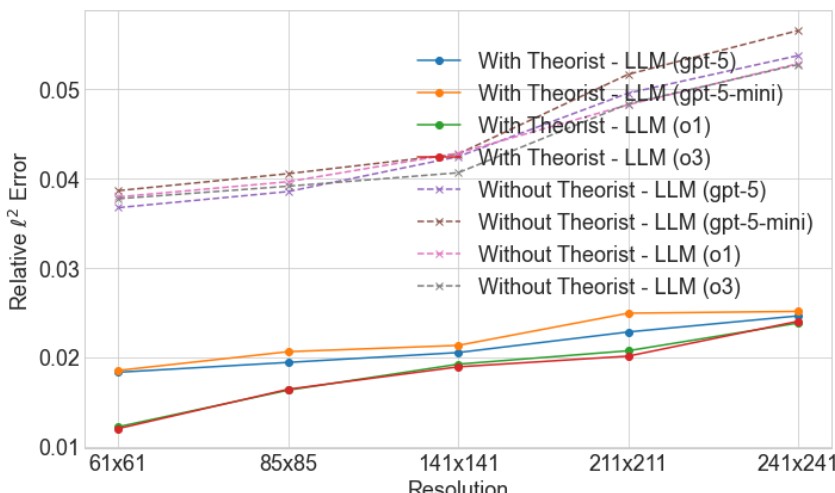

Figure 2: Relative $\ell^2$ error comparisons of neural operators designed by LLM frameworks with and without Theorist on different resolutions.

theoretical insights, guarantees that the neural operators exhibit strong generalization capability with respect to the number of sampling points.

### 4.3 CAN CRITIC AND REFINER PRODUCE BETTER RESULTS?

In our framework, collaboration between the Critic and Refiner to identify drawbacks and potential issues in both the theory and the code implementation is a key step toward improving mathematical soundness, performance, and generalization. The Refiner step has been demonstrated to possess strong debugging capability (Li et al., 2025). To analyze the contributions of the Critic and Refiner steps, we compare our full framework to a variant that includes only the Refiner step, which revises code according to reported bugs. Without the Critic step, we hypothesize the following:

> **Hypothesis**
>
> The quality of the output of Theorist directly decides the performance of the LLM-designed neural operators.

We then conduct experiments on the Darcy dataset and evaluate the relative $\ell^2$ error across different resolutions. Moreover, we introduce another LLM (Gemini 2.0 Thinking) as a judge, assigning a score (with a full mark of 5) to quantify the quality of the Theorist's feedback.

| Model | $61 \times 61$ | $85 \times 85$ | $141 \times 141$ | $211 \times 211$ | $241 \times 241$ | Score |
|---|---|---|---|---|---|---|
| **With Critic** | | | | | | |
| LLM (gpt-5) | 0.0183 | 0.0194 | 0.0205 | 0.0228 | 0.0246 | 4.5 |
| LLM (gpt-5-mini) | 0.0185 | 0.0206 | 0.0213 | 0.0249 | 0.0251 | 4.4 |
| LLM (o1) | 0.0122 | 0.0163 | 0.0192 | 0.0207 | 0.0238 | 4.6 |
| LLM (o3) | 0.0120 | 0.0164 | 0.0189 | 0.0201 | 0.0240 | 4.6 |
| **Without Critic** | | | | | | |
| LLM (gpt-5) | 0.0191 | 0.0216 | 0.0224 | 0.0231 | 0.0259 | 4.1 |
| LLM (gpt-5-mini) | 0.0305 | 0.0341 | 0.0355 | 0.0368 | 0.0384 | 3.5 |
| LLM (o1) | 0.0160 | 0.0177 | 0.0201 | 0.0216 | 0.0249 | 4.4 |
| LLM (o3) | 0.0162 | 0.0176 | 0.0204 | 0.0218 | 0.0257 | 4.4 |

Table 3: Relative $\ell^2$ error and score comparisons of neural operators designed by LLM frameworks with and without Critic across six benchmark datasets.

Table 3 reports the relative $\ell^2$ errors of neural operators designed without the Critic step. For `gpt-5`, `o1`, and `o3`, the Theorist provides high-quality feedback, resulting in only a slight decrease in performance. However, for `gpt-5-mini`, the performance drops by more than $30\%$.

## 4.4 CAN LLMs DESIGN NEURAL OPERATORS USING OBSCURE MATH?

One key observation during the design process is stated as follows:

> **Observation**
>
> Theorist tends to generalize the well-established theoretical insights and incorporate them into the selected backbone.

Although the LLM-designed neural operators are novel, we further analyze whether LLMs can design neural operators based on obscure mathematical results, thereby extending the boundary of neural operators from combinations of existing components and incremental contributions to existing architectures to fully new neural operators first proposed in the literature. Obscure mathematical results refer to theories that have attracted attention from only a small fraction of researchers worldwide. Given the lack of training data on obscure mathematics, LLMs may struggle with these topics. Therefore, we conduct a new ablation study that instructs the Theorist to design a neural operator utilizing adaptive Fourier decomposition (AFD) (Qian et al., 2011) on the 1D Burgers' equation dataset (Li et al., 2020b). We find that the relative $\ell^2$ errors of neural operators guided by AFD are consistently one to two orders of magnitude lower than those of neural operators designed using mainstream theories across all four LLM models. For example, with GPT-5, the LLM-designed AFD neural operator attains a relative $\ell^2$ error of $0.013$, whereas the LLM-designed neural operator based on FNO achieves $0.0006$.

In contrast, we also design a neural operator based on AFD ourselves, which reaches a relative $\ell^2$ error of $0.0009$. That said, AFD has the potential to be integrated into neural operators to improve their performance. However, LLM-designed AFD neural operators perform worse than the human-designed version. Ultimately, we find that the reason is that the Theorist does not fully understand AFD. To illustrate this, we showcase the *Choice and justification* section of the Theorist's feedback, where red indicates the parts of AFD understood by the Theorist:

> Choice and justification
> - Choose an adaptive Fourier decomposition-based neural operator (AFD-NO): a Fourier neural operator in space with an adaptive, data-dependent selection/mixing of Fourier modes, wrapped inside a stable exponential time-differencing (ETD-1) integrator for the viscous term.
> - Why it is best here:
> 1) Periodic boundary conditions and low-frequency initial conditions make the Fourier basis the natural eigenbasis; Burgers' nonlinearity redistributes energy across modes, so learning in the spectral domain is efficient.
> 2) Adaptivity is key: noisy initial data and the incipient formation of steep gradients require selectively attending to and mixing the subset of active modes. AFD layers learn soft, sample-dependent spectral masks and frequency-wise linear maps, improving data efficiency and robustness versus fixed-mode FNOs.
> 3) Stability and inductive bias: we treat the viscous linear part exactly in Fourier (ETD-1), leaving the network to learn only the convective nonlinearity. This aligns with the PDE's semigroup structure and improves numerical stability for $\mu = 0.01$.

With domain knowledge of AFD, this statement has been evaluated by several senior researchers in the field. It was found that LLMs do not grasp the essence of AFD, which is the rational approximation via pole selection in a reproducing kernel Hilbert space. Instead, LLMs conflate Fourier decomposition with the Fourier transform and interpret poles as active modes. Additionally, they ignore the requirements on the basis and the function space needed to implement AFD. Overall, LLMs produce misleading and incorrect content due to hallucination. Moreover, LLMs tend to overfit to similar terms involving adaptivity in Fourier transform theory, overlooking their differences.

We point out that our prompting strategy is highly structured to constrain the LLM's reasoning space, rather than relying on it to invent mathematics from scratch. Specifically, the prompt first explicitly provides the Theorist with a *factory of existing neural operators* (including FNO, DeepONet, LNO, etc.) as an in-context "toolbox". It then instructs it to first select the most appropriate backbone architecture from this toolbox. Next, it guides it to propose a specific mathematical modification to align that backbone with the specific theoretical properties of the given PDE (such as stiffness, boundary conditions, or conservation laws). Meanwhile, the AFD failure case validates this strategy: when the LLM was forced to use an "obscure" theory not in its pre-trained knowledge base, it began to "hallucinate" and conflate concepts. Therefore, our framework's success relies on guiding the LLM to apply theories it already understands well and are validated, not on its ability to perform novel or obscure mathematical derivations.

Furthermore, it is worth noting that the ablation study presented in Section 4.2 only shows the impact of the Theorist component, not its initial correctness. The mechanism we used for the independent validation of the Theorist's output was the human expert review. To clarify, this review was not conducted after the Critic or Refiner intervened. Instead, it was performed specifically on the initial mathematical formulation and architecture description generated by the Theorist, before they were passed to the Critic. For 5 of the 6 benchmark problems we tested, the Theorist's initial theoretical proposal was judged "Yes" (i.e., theoretically correct and logically sound) by the human experts. This indicates that the Theorist provided a solid theoretical foundation in the majority of cases. Subsequently, the Critic's role was focused on identifying implementation-level issues (such as numerical instability, code inefficiency, or edge cases) rather than correcting fundamental theoretical errors. The only exception was the AFD case, where the initial theory was indeed flawed, and this was accurately identified by our human expert review. This confirms that the Theorist's output is, by and large, theoretically reliable before entering the Critic's review loop.

### 4.5 HOW MUCH TIME DOES IT TAKE FOR LLM TO DESIGN A NEURAL OPERATOR?

In terms of design time cost, for a typical benchmark problem (e.g., 2D Navier-Stokes), the average wall-clock time for our four-agent framework to go from receiving the problem specification to generating a validated, bug-free final operator code is about 35-45 minutes. This process, running on our experimental workstation (equipped with a single NVIDIA RTX 4090), requires an average of 7-9 full agent iterations (i.e., the Theorist → Programmer → Critic → Refiner loop).

While the "human expert effort" baseline is difficult to quantify precisely, it is known that neural operator design is still considered more of an art than a science (Sanderse et al., 2025), typically involving deep intuition, expert experience, numerous iterations, and trial-and-error experimentation. Thus, our proposed automated process represents a significant compression in time cost compared to the days or even weeks required for a human expert to design, implement, and debug a novel, competitive neural operator architecture. Furthermore, it is worth mentioning that the time it takes for a non-expert in neural operators to develop a working neural operator solver for PDEs will be much longer.

## 5 RELATED WORK

**Neural operators.** Operator learning targets mappings between infinite-dimensional function spaces, enabling resolution- and mesh-independent surrogates for PDE families. Two foundational approaches are DeepONet (Lu et al., 2019), which learns nonlinear operators via a branch-trunk factorization, and the Fourier Neural Operator (FNO) (Li et al., 2020b), which applies spectral convolutions to achieve strong resolution transfer on canonical elliptic/parabolic problems. Subsequent variants improve accuracy, efficiency, or inductive bias: U-FNO couples Fourier mixing with U-Net refinements (Wen et al., 2022); F-FNO factorizes spectral weight tensors to lower complexity (Tran et al., 2021); multiwavelet formulations provide multiresolution locality and sparsity (Gupta et al., 2021; 2022; Tripura & Chakraborty, 2023); ONO augments operator learning with orthogonalized kernels and stability enhancements (Xiao et al., 2023); Galerkin operators embed variational structure (Cao, 2021); LSM exploits learned spectral methods (Wu et al., 2023); and transformer-style operators (OFormer, Transolver) leverage attention for long-range coupling (Li et al., 2022; Wu et al., 2024). Latent designs (LNO, LaMO) compress fields into compact representations to balance accuracy and cost (Wang & Wang, 2024; Tiwari et al., 2025). Beyond periodic grids, irregular ge-

ometries and structured meshes (e.g., airfoil, plasticity, pipe, elasticity) stress resolution transfer and generalization and have motivated the architectural choices and benchmarks used in this work (Li et al., 2023). Our study differs by asking whether an LLM can *select or synthesize* such operator ingredients end-to-end, guided by mathematical analysis rather than manual trial-and-error.

**LLMs in scientific machine learning.** LLMs have been used to automate elements of scientific workflows: program synthesis and code repair (Koziolek et al., 2024; Xu et al., 2024a), robotics/mobile task automation (Wen et al., 2024; Guan et al., 2024), hardware and systems design (Xu et al., 2024b), scientific discovery pipelines (Zimmermann et al., 2025; Filimonov, 2024; Aamer et al., 2025), hyperparameter tuning (Zhang et al., 2023), and mathematical problem solving (Bian et al., 2025). For PDEs, hybrid methods insert LLMs as components within neural architectures or to provide natural-language rationales, but stop short of automating the full design loop (Zhou et al., 2025; Lorsung & Farimani, 2024). Closer to our goal are fully automated agents that generate solvers: CodePDE synthesizes and evaluates traditional PDE codes (Li et al., 2025), and PINNsAgent targets PINN implementations (Wuwu et al., 2025). However, these systems optimize numerical performance without enforcing operator-theoretic design principles. In contrast, we position the LLM as a theory-aware designer that proposes and justifies operator choices (e.g., spectral vs. multiresolution vs. latent), then compiles them into executable PyTorch implementations subject to physics and numerical checks.

**Automated agent systems.** Role specialization and multi-agent coordination have been shown to improve complex code generation and iterative refinement via planning, self-critique, and division of labor (Carlander et al., 2024; Dong et al., 2024; Takagi et al., 2025). Recent agentic SciML systems instantiate plan-execute-review loops for PDE tasks but largely emphasize execution or empirical tuning (Li et al., 2025; Wuwu et al., 2025). Our pipeline adopts a four-role decomposition, *Theorist* (formal derivation and architectural justification), *Programmer* (faithful implementation), *Critic* (adversarial numerical/software review), and *Refiner* (targeted fixes), explicitly coupling mathematical validation with software iteration. This theory-aware agent design aims to reduce hallucinations, improve stability under discretization changes, and systematize operator selection, compared with prior agent frameworks that lack principled operator-level guidance.

## 6 CONCLUSION

We asked whether large language models can design neural operators that are accurate, efficient, and robust, grounded not only in empirical performance but also in mathematical reasoning. To this end, we introduced a four-agent pipeline that closes the loop from problem specification and theory selection to implementation, adversarial review, refinement, and automatic training/evaluation. Across six PDE benchmarks, the LLM-designed operators consistently matched or outperformed strong human-designed baselines in relative $\ell^2$ error while using fewer parameters and exhibiting favorable training-time and memory profiles. The core of the LLM's success lies in its ability to expertly consolidate and apply well-established mathematical concepts under the rigorous guidance of the multi-agent system. The LLM's training data provides a strong, latent "code-to-math" and "math-to-code" inductive bias. Its ability to translate high-level mathematical concepts (like Fourier bases, convolution, or spectral methods) into syntactically correct and structurally efficient PyTorch modules poses a significant advantage over humans. Taken together, neural-operator design can be systematized as a coupled process of theory, implementation, and structured review, moving the task from "art" toward "science".

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
