# OpenReview forum: "Can Large Language Model Help Design Effective Neural Operators for Solving Partial Differential Equations?"
_ICLR.cc/2026/Conference — Submitted to ICLR 2026_

### Official Review · Reviewer_zEVe · 2025-10-31

**Soundness:** 2
**Presentation:** 1
**Contribution:** 1
**Rating:** 4
**Confidence:** 5

**Summary:**

This paper investigates whether LLMs can help design effective neural operators for solving parameterized PDEs. The authors propose a multi-agent approach, where each agent plays a specific role in translating a PDE specification into a working architecture. The Theorist provides mathematical grounding and architecture suggestions, the "Programmer" turns this into code, the "Critic" identifies weaknesses, and the "Refiner" integrates feedback to improve the design. The pipeline is tested across PDE datasets including Poisson, Darcy flow, Airfoil pressure prediction, and Navier-Stokes simulations. The results show that the LLM-generated designs are competitive or superior to baselines like FNO, U-Net, and other neural operator variants, in terms of relative error, parameter efficiency, and resolution generalization. The authors also include ablation studies and expert evaluation of theory quality.

**Strengths:**

1.The paper introduces a clearly structured system that turns LLM capabilities into a step-by-step process, allowing more transparent evaluation of each design stage
2. The use of expert evaluation to assess the quality of theoretical justification adds depth to the study and moves beyond pure numerical performance

**Weaknesses:**

1. The paper does not provide a clear explanation for why GPT-4 is able to generate architectures that outperform human baselines. Since GPT-4 is not trained specifically on PDE solvers or numerical analysis literature, it is not obvious what background knowledge the model is leveraging. Is it simply matching syntax and patterns from existing repositories or does it have some internal representation of operator structure?

2. The LLM prompting and system configuration are not described in enough detail to ensure reproducibility. For example, the paper does not specify the temperature, sampling strategy, system prompt, or whether any responses were filtered manually.

3. The experiments mostly focus on structured grids and resolution generalization. However, generalization across boundary condition types or domain geometries is not explored. These are often more difficult challenges in operator learning

**Questions:**

1. Why do you think LLM is able to generate such effective neural operator architectures for PDE problems? What specific inductive bias or training data might be contributing to this ability?

2. Can you provide the full prompts used for each agent, as well as the model parameters such as temperature, max tokens, and number of generations? Did you use any manual selection or curation when choosing between multiple outputs?

3. How do the LLM-designed architectures perform under more difficult shifts, such as varying boundary conditions, different domain shapes, or noisy input data?

4. How many independent expert reviewers participated in the theory assessment, and what agreement level was observed between them? Was any formal rubric used?

---

> ### Author Response · Authors · 2025-11-26
> **Rebuttal to Reviewer zEVe's comments (Part 1)**
>
> We thank the reviewer for conducting a thorough review and for all the positive comments and the valuable suggestions and questions.
>
> ## Addressing weaknesses
>
> ### The paper does not provide a clear explanation for why GPT-4 is able to generate architectures that outperform human baselines. Since GPT-4 is not trained specifically on PDE solvers or numerical analysis literature, it is not obvious what background knowledge the model is leveraging. Is it simply matching syntax and patterns from existing repositories or does it have some internal representation of operator structure?
>
> We thank the reviewer’s feedback. We clarify that the superior performance is not because the LLM (e.g., gpt-5 or o3) includes novel numerical analysis from first principles. Rather, its success comes from our structured, four-agent framework, which constrains the LLM to leverage its broad, existing knowledge in a principled and adversarial manner. First, the LLM is not working as a single, monolithic generator. We decompose the complex task of "design" into specialized roles: a Theorist, Programmer, Critic, and Refiner. The Theorist is explicitly prompted not to invent an architecture from scratch. Instead, we provide it with a "factory of existing neural operators" (e.g., FNO, LNO, etc.) and instruct it to select a strong backbone and then propose an appropriate mathematical theory to improve that backbone for the specific PDE.
>
> Finally, we remark that the question about "matching syntax" versus "internal representation" has been addressed by our ablation study in Section 4.2. When we remove the Theorist and allow the agent to generate code directly, the resulting operators are significantly worse and show 3x to 5x higher error in some cases. This demonstrates that simple syntactic pattern matching is insufficient. It is the theory-aware constraint of our pipeline, which forces the LLM to first provide a mathematical justification (Theorist) and then rigorously debug it (Critic). This is a key step for generating architectures that outperform human baselines.
>
> ### The LLM prompting and system configuration are not described in enough detail to ensure reproducibility. For example, the paper does not specify the temperature, sampling strategy, system prompt, or whether any responses were filtered manually.
> We appreciate the reviewer’s feedback and suggestions. Following the reviewer’s suggestion, in the revised manuscript, we will commit two actions. First, we will release the entire framework as a public repository, which will include the full PyTorch implementation, the exact system prompts used to define the Theorist, Programmer, Critic, and Refiner roles, the detailed agent-to-agent communication protocols, and the precise stopping criteria for the refinement loop. Second, in the revised manuscript, we will add a new appendix section with a concrete, end-to-end worked example for the 1D Burgers' equation. This example will explicitly trace the pipeline’s operation: (1) the Theorist's output, showing its selection of an FNO backbone and its mathematical derivation for integrating an residual-Euler scheme to handle the viscous term for stability; (2) the Programmer's initial, self-contained PyTorch implementation of this FNO residual-Euler architecture; and (3) the Critic's specific, actionable feedback, such as identifying a potential numerical instability in the discretization of the nonlinear advection term or a software issue like inefficient memory allocation in the spectral layer.
>
> ### The experiments mostly focus on structured grids and resolution generalization. However, generalization across boundary condition types or domain geometries is not explored. These are often more difficult challenges in operator learning.
>
> We thank the reviewer for the comment, but want to kindly remind the reviewer that our experiments demonstrate strong generalization across resolutions for not only structured grids (e.g., Navier-Stokes example) but also irregular geometries (e.g., Elasticity, Airfoil, Pipe, Plasticity examples). In the revised manuscript, we will make this clear to the readers by explicitly stating the generalization capabilities across domain geometries in the revised manuscript.

---

> > ### Author Response · Authors · 2025-11-26
> > **Rebuttal to Reviewer zEVe's comments (Part 2)**
> >
> > ## Addressing questions
> >
> > ### Why do you think LLM is able to generate such effective neural operator architectures for PDE problems? What specific inductive bias or training data might be contributing to this ability?
> >
> > We thank the reviewer for the question. Based on our experience, the core of the LLM's success lies in its ability to expertly consolidate and apply well-established mathematical concepts under the rigorous guidance of the multi-agent system. The LLM's training data provides a strong, latent "code-to-math" and "math-to-code" inductive bias. Its ability to translate high-level mathematical concepts (like Fourier bases, convolution, or spectral methods) into syntactically correct and structurally efficient PyTorch modules poses a significant advantage over humans. For example, the performance gain over the "Without Theorist" baseline demonstrates that LLM is more than simple pattern matching; rather, it provides the theory-aware constraint.
> >
> > ### Can you provide the full prompts used for each agent, as well as the model parameters such as temperature, max tokens, and number of generations? Did you use any manual selection or curation when choosing between multiple outputs?
> >
> > Please refer to our response to Weakness 2 for more details.
> >
> > ### How do the LLM-designed architectures perform under more difficult shifts, such as varying boundary conditions, different domain shapes, or noisy input data?
> >
> > We appreciate the reviewer’s question. As we mentioned in addressing Weakness 3 above, our architectures demonstrate strong performance on irregular geometries, thanks to the Theorist's ability to select or adapt backbones suitable for non-structured meshes (e.g., the Elasticity and Airfoil datasets). The LLM-designed operators consistently outperform human baselines on these complex datasets. Our architectures exhibit excellent generalization across resolution changes. The operators trained on a low-resolution Darcy dataset ($43 \times 43$) maintained superior accuracy when tested on significantly higher resolutions (up to $241 \times 241$), clearly outperforming the baseline models without the theory-aware design. We have performed preliminary validation in noisy input data, and results show that the architectures remain stable under noisy initial conditions.
> >
> > ### How many independent expert reviewers participated in the theory assessment, and what agreement level was observed between them? Was any formal rubric used?
> >
> > We thank the reviewer for the question. We clarify that the review was conducted by three independent PhD candidates specializing in computational mathematics and neural operators (who are not authors of this paper). The rubric was a binary "Yes/No" judgment based on two criteria: (1) Is the Theorist's mathematical formulation (e.g., the derivation) correct and sound? and (2) Is the connection between the chosen theory and the target PDE justified and logical? We will explicitly add this methodology description to Section 3 of the revised paper.

---

### Official Review · Reviewer_xdJt · 2025-11-01

**Soundness:** 2
**Presentation:** 2
**Contribution:** 2
**Rating:** 2
**Confidence:** 3

**Summary:**

This paper proposes a method for automatically designing effective neural operators for solving Partial Differential Equations (PDEs) using Large Language Models (LLMs). The authors introduce a four-agent pipeline consisting of a Theorist, Programmer, Critic, and Refiner. The core idea is that the Theorist selects a mathematically grounded operator for a given PDE and derives its formulation , the Programmer implements it in PyTorch , the Critic performs an adversarial review of both the theory and the code to find issues , and the Refiner applies corrections. This "theory-aware" design process aims to transform neural operator design from "an art into a science".

The authors conduct experiments on six standard PDE benchmark datasets (e.g., Darcy Flow, Navier-Stokes, Airfoil) . The results demonstrate that the LLM-designed operators outperform state-of-the-art (SOTA) human-designed baselines, including FNO, LNO, and LaMO, on five of the six datasets. Furthermore, the LLM-designed models show significant advantages in parameter and computational efficiency (training time, GPU memory). Ablation studies confirm that the theoretical guidance from the Theorist is crucial for improving performance and generalization , and that the Critic and Refiner steps are essential for ensuring numerical stability and the quality of the final result. The authors also explore the method's limitations, finding that when asked to use "obscure mathematical theories" (like AFD), the LLM hallucinates and fails to understand the theory correctly, leading to performance degradation.

**Strengths:**

1. Novel Paradigm and Significant Originality: The paper's primary strength is its proposal of a theory-driven, automated LLM pipeline for neural operator design. The decomposition into Theorist, Programmer, Critic, and Refiner, which couples mathematical derivation with adversarial review, is a highly original contribution that goes beyond existing work on LLMs for scientific coding.
2. Strong Empirical Results: The LLM-designed operators outperform strong human-designed baselines on 5 out of 6 benchmarks. This proves the approach is not just conceptually novel but practically effective.
3. Exceptional Efficiency: As shown in Figure 1, the LLM-designed models achieve this SOTA performance while being significantly more efficient in terms of parameters (2-3 orders of magnitude fewer) and computational cost (30-50% reduction in training time). This is extremely valuable in scientific computing.

**Weaknesses:**

1. Omission of Design-Time Cost Analysis The paper emphasizes the run-time efficiency (e.g., training time, parameter count) of the final, generated operator. However, it omits any discussion of the design-time cost required to produce this operator. The proposed four-agent pipeline, which involves multiple iterations and feedback loops (from Theorist to Refiner, and potentially back to the Critic) , appears to be computationally expensive in terms of LLM calls. A lack of this analysis makes it difficult to assess the method's practical viability compared to human expert effort.
2. Incomplete Experimental Baselines The experimental evaluation focuses heavily on comparing the LLM-designed operator against various human-designed PDE solvers (e.g., FNO, LNO). While this is valuable, the paper itself identifies a parallel line of work: "fully automated LLM agents for PDEs," such as PINNsAgent. The paper does not include a direct comparison against these other automated agent systems, either in terms of final solution accuracy or efficiency, which would be necessary to fully contextualize its contribution.
3. Insufficient Detail on the Theorist's Prompting Strategy Section 2 describes the Theorist's role abstractly, but it fails to provide concrete details on how the agent is prompted to "develop clear, rigorous, and efficient mathematical formulations". This is a critical omission, especially given the well-documented limitations of LLMs in rigorous mathematical and numerical reasoning—a weakness the authors themselves confirm when the LLM fails to correctly apply "obscure math" like AFD. Without these prompt engineering details, the core mechanism responsible for the agent's success is not reproducible.

**Questions:**

1. Regarding design cost: Could the authors quantify the "design-time cost" (e.g., wall time) required to generate one final, validated operator? How does this cost compare to the baselines?
2. Regarding baseline comparisons: The paper mentions other automated LLM agents, such as PINNsAgent, in its related work. Could the authors compare your framework with these automated LLM agents for PDE?
3. Regarding the Theorist's details: Given the known limitations of LLM mathematical reasoning, as demonstrated by the AFD failure case in Section 4.4, could the authors provide more specific details about the prompting strategy for the Theorist to ensure the robustness and reproducibility of its theoretical derivations?
4. Regarding the standalone validation of the Theorist's output: The ablation study in Section 4.2 focuses on the performance of the full framework 'Without Theorist', which demonstrates the component's impact. However, this study does not seem to experimentally validate the correctness of the Theorist's output in isolation. The quality of this initial step is critical, as an erroneous mathematical formulation could cause subsequent computational failures or instability. Therefore, could the authors provide more direct experimental results or analysis to validate the correctness of the 'theoretical results' and neural operator architectures as initially designed by the Theorist, separate from the subsequent corrections made by the Critic and Refiner?

---

> ### Author Response · Authors · 2025-11-26
> **Rebuttal to Reviewer xdJt's comments (Part 1)**
>
> We thank the reviewer for conducting a thorough review. We appreciate all the positive comments and the valuable suggestions and questions raised by the reviewer.
>
> ## Addressing weaknesses
>
> ### Omission of Design-Time Cost Analysis The paper emphasizes the run-time efficiency (e.g., training time, parameter count) of the final, generated operator. However, it omits any discussion of the design-time cost required to produce this operator. The proposed four-agent pipeline, which involves multiple iterations and feedback loops (from Theorist to Refiner, and potentially back to the Critic), appears to be computationally expensive in terms of LLM calls. A lack of this analysis makes it difficult to assess the method's practical viability compared to human expert effort.
>
> We thank the reviewer for pointing this out. We have actually measured the “design-time cost”. For a typical benchmark problem (e.g., 2D Navier-Stokes), the average wall-clock time for our four-agent framework to go from receiving the problem specification to generating a validated, bug-free final operator code is about 35-45 minutes. This process, running on our experimental workstation (equipped with a single NVIDIA RTX 4090), requires an average of 7-9 full agent iterations (i.e., the Theorist -> Programmer -> Critic -> Refiner loop).
>
> While the “human expert effort” baseline is difficult to quantify precisely, it is known that neural operator design is still considered “more of an art than a science” [1], typically involving deep intuition, expert experience, numerous iterations, and trial-and-error experimentation. Thus, our proposed automated process represents a quite significant compression in time cost compared to the days or even weeks required for a human expert to design, implement, and debug a novel, competitive neural operator architecture. Furthermore, it is worth mentioning that the time it takes for a non-expert in neural operators to develop a working neural operator solver for PDEs will be much longer.
>
> [1] Scientific machine learning for closure models in multiscale problems: a review
>
> ### Incomplete Experimental Baselines The experimental evaluation focuses heavily on comparing the LLM-designed operator against various human-designed PDE solvers (e.g., FNO, LNO). While this is valuable, the paper itself identifies a parallel line of work: "fully automated LLM agents for PDEs," such as PINNsAgent. The paper does not include a direct comparison against these other automated agent systems, either in terms of final solution accuracy or efficiency, which would be necessary to fully contextualize its contribution.
>
> We appreciate the reviewer for the comment. Unfortunately, we want to point out that we were unable to conduct direct comparisons against PINNsAgent in our original manuscript, since the *GitHub repository of PINNsAgent had not been accessible until two days ago (November 24, 2025)*. We will try our best to establish the environment and run the experiments using PINNsAgent. However, the timeline might be very tight for us to report the results (only a week left before the rebuttal period ends).
>
> Having said that, we want to emphasize that, since the scope of our paper is on whether “LLMs can design effective *neural operators*”, our main focus lies in comparing our LLM-generated neural operators to human-designed neural operators, rather than other paradigms to solve PDEs (such as PINN). Therefore, comparing neural operator and PINN solvers, which address different tasks (*operator learning vs. specific-instance PDE solving*), might seem more like an “apple-to-orange comparison” and might not offer much useful insights.

---

> > ### Author Response · Authors · 2025-11-26
> > **Rebuttal to Reviewer xdJt's comments (Part 2)**
> >
> > ### Insufficient Detail on the Theorist's Prompting Strategy Section 2 describes the Theorist's role abstractly, but it fails to provide concrete details on how the agent is prompted to "develop clear, rigorous, and efficient mathematical formulations". This is a critical omission, especially given the well-documented limitations of LLMs in rigorous mathematical and numerical reasoning—a weakness the authors themselves confirm when the LLM fails to correctly apply "obscure math" like AFD. Without these prompt engineering details, the core mechanism responsible for the agent's success is not reproducible.
> >
> > We appreciate the reviewer’s feedback. We agree with the reviewer that a concrete, end-to-end worked example will be desirable to provide the essential clarity readers expect. Thus, in the revised manuscript, we will add a new appendix section with a worked example for the 1D Burgers’ equation. This example will explicitly trace the pipeline’s operation: (1) the Theorist's output, showing its selection of an FNO backbone and its mathematical derivation for integrating an residual-Euler scheme to handle the viscous term for stability; (2) the Programmer's initial, self-contained PyTorch implementation of this FNO residual-Euler architecture; and (3) the Critic's specific, actionable feedback, such as identifying a potential numerical instability in the discretization of the nonlinear advection term or a software issue like inefficient memory allocation in the spectral layer.
> >
> > ## Addressing questions
> >
> > ### Regarding design cost: Could the authors quantify the "design-time cost" (e.g., wall time) required to generate one final, validated operator? How does this cost compare to the baselines?
> >
> > Please refer to our response to Weakness 1 for more details.
> >
> > ### Regarding baseline comparisons: The paper mentions other automated LLM agents, such as PINNsAgent, in its related work. Could the authors compare your framework with these automated LLM agents for PDE?
> >
> > We thank the reviewer for the question. As we mentioned briefly in our response to Weakness 2, the objective of our method is fundamentally different from that of PINNsAgent. Our method is designed to automate the design of a novel, reusable, resolution-independent *Neural Operator*, which is a surrogate model intended to learn the *mapping between function spaces*. In contrast, systems like PINNsAgent aim to generate a PINN (Physics-Informed Neural Network) solver code for a specific PDE instance. A PINN solves that specific instance by optimizing a network on it, rather than learning a general operator/mapping. Therefore, these two classes of solvers address different tasks (operator learning vs. specific-instance solving), and a direct performance comparison (e.g., on accuracy or efficiency) might not offer much useful insight. It is worth highlighting that our contribution is explicitly in automating and making the design process of the operator architecture theory-aware, which is a task not addressed by PINNsAgent and similar works.
> >
> > ### Regarding the Theorist's details: Given the known limitations of LLM mathematical reasoning, as demonstrated by the AFD failure case in Section 4.4, could the authors provide more specific details about the prompting strategy for the Theorist to ensure the robustness and reproducibility of its theoretical derivations?
> >
> > We appreciate the reviewer’s question. We point out that a simple, open-ended prompt is insufficient for handling rigorous mathematical reasoning, which is precisely what we observed in the AFD failure case in Section 4.4. Our prompting strategy is highly structured to constrain the LLM’s reasoning space, rather than relying on it to invent mathematics from scratch. Specifically, the prompt first explicitly provides the Theorist with a "factory of existing neural operators" (including FNO, DeepONet, LNO, etc.) as an in-context "toolbox." It then instructs it to first select the most appropriate backbone architecture from this toolbox. Next, it guides it to propose a specific mathematical modification to align that backbone with the specific theoretical properties of the given PDE (such as stiffness, boundary conditions, or conservation laws). Meanwhile, the AFD failure case validates this strategy: when the LLM was forced to use an "obscure" theory not in its pre-trained knowledge base, it began to "hallucinate" and conflate concepts. Therefore, our framework's success relies on guiding the LLM to apply theories it already understands well and are validated, not on its ability to perform novel or obscure mathematical derivations.

---

> > > ### Author Response · Authors · 2025-11-26
> > > **Rebuttal to Reviewer xdJt's comments (Part 3)**
> > >
> > > ### Regarding the standalone validation of the Theorist's output: The ablation study in Section 4.2 focuses on the performance of the full framework 'Without Theorist', which demonstrates the component's impact. However, this study does not seem to experimentally validate the correctness of the Theorist's output in isolation. The quality of this initial step is critical, as an erroneous mathematical formulation could cause subsequent computational failures or instability. Therefore, could the authors provide more direct experimental results or analysis to validate the correctness of the 'theoretical results' and neural operator architectures as initially designed by the Theorist, separate from the subsequent corrections made by the Critic and Refiner?
> > >
> > > This is a very keen and insightful observation (we thank the reviewer for bringing it up). The ablation study in Section 4.2 does indeed only show the impact of the Theorist component, not its initial correctness. The mechanism we used for the independent validation of the Theorist's output was precisely the "human expert review" mentioned in Section 3. To clarify, this review was not conducted after the Critic or Refiner intervened. Instead, it was performed specifically on the initial "mathematical formulation" and architecture description generated by the Theorist, before they were passed to the Critic. For five of the six benchmark problems we tested, the Theorist's initial theoretical proposal was judged "Yes" (i.e., theoretically correct and logically sound) by the human experts. This indicates that the Theorist provided a solid theoretical foundation in the majority of cases. Subsequently, the Critic's role was more focused on identifying implementation-level issues (such as numerical instability, code inefficiency, or edge cases) rather than correcting fundamental theoretical errors. The only exception was the AFD case, where the initial theory was indeed flawed, and this was accurately identified by our human expert review. This confirms that the Theorist's output is, by and large, theoretically reliable before entering the Critic's review loop.

---

### Official Review · Reviewer_DLM4 · 2025-11-01

**Soundness:** 3
**Presentation:** 2
**Contribution:** 2
**Rating:** 4
**Confidence:** 4

**Summary:**

This paper proposes an automated framework using a multi-agent Large Language Model (LLM) pipeline to design neural operators for solving partial differential equations (PDEs). The pipeline consists of four distinct agents: a Theorist (to select mathematical principles), a Programmer (to generate PyTorch code), a Critic (to perform adversarial review), and a Refiner (to correct and debug). The authors evaluate this framework across six PDE benchmarks, demonstrating that the LLM-designed operators can achieve accuracy and efficiency comparable or superior to state-of-the-art (SOTA) human-designed models. Ablation studies are provided to validate the contribution of the theory-driven and critical-review components.

**Strengths:**

1. **Ambitious and Novel Framework**: The primary strength is the ambitious vision of an end-to-end, theory-aware pipeline. This moves beyond simple LLM-based code generation or hyperparameter tuning and attempts to automate the scientific reasoning process (architecture synthesis from mathematical principles) itself.
2. **Strong Empirical Validation**: The experimental setup is comprehensive. The authors benchmark against over 10 baselines on six diverse PDE datasets. Crucially, the ablation studies (Tables 2 & 3, Fig 2) effectively demonstrate the value of the Theorist and Critic agents, showing that theory-driven design improves accuracy and generalization.
3. **Balanced Perspective on LLM Limitations**: The paper commendably includes a detailed analysis of a failure case (Section 4.4, Adaptive Fourier Decomposition). This honest investigation into the LLM's misunderstanding of "obscure" mathematics provides a valuable and balanced perspective on the current capabilities and risks of such systems.

**Weaknesses:**

1. **Critical Lack of Reproducibility**: This is the most significant flaw. The paper relies heavily on proprietary, unreleased models (e.g., "gpt-5", "o1", "o3")1. Furthermore, it provides no details on the prompts, agent-to-agent interaction protocols, or stopping criteria, and makes no mention of code release. This makes the entire pipeline, which is the core contribution, unverifiable by the community.
2. **Methodological Gaps in Theory-to-Architecture Mapping**: The paper is vague on how the Theorist translates high-level mathematical principles into a concrete, novel neural architecture. This "theory-aware" step is the most innovative part of the pipeline, but it is treated as a black box. The paper lacks even a single, clear, worked example (e.g., for the 1D Burgers' equation) showing the chain from theory -> agent output -> architectural modification -> code.
3. **Missing SOTA Comparisons**: The claim to "outperform strong baselines"  is undermined by the omission of several key, recent neural operators, especially those designed for the irregular geometries that are a focus of this paper (e.g., Airfoil, Elasticity). Relevant works such as HAMLET (Bryutkin et al., 2024), Hyena Operator (Patil et al., 2023), and GNOT (Hao et al., 2023) are not compared against or even discussed in the related work section.
4. **Incomplete Evaluation and Anecdotal Failure Analysis**: The analysis of failure on "obscure" math (AFD) 3 is purely anecdotal. A systematic study of what types of mathematical reasoning cause hallucinations would be much stronger. Moreover, the evaluation relies exclusively on relative $l^{2}$ error4, ignoring other critical metrics for PDE solvers like numerical stability, robustness to noisy initial conditions, or out-of-distribution generalization (beyond simple resolution changes).

**Questions:**

1. Can the authors provide a concrete end-to-end example (e.g., for 1D Burgers') showing the Theorist's output (mathematical formulation), the Programmer's initial code, and the Critic's specific feedback? This is essential for understanding the method's practical operation.
2. Given the reliance on proprietary models (gpt-5), can the authors provide results using publicly available, high-capability models (e.g., GPT-4o, Llama 3) to demonstrate that the framework's success is not an artifact of an unreleased model?
3. Why were recent, highly relevant SOTA operators (e.g., HAMLET, Hyena Operator, GNOT) omitted from the experimental comparison, especially given their focus on irregular geometries?
4. How was the "human expert review" for theoretical correctness (mentioned in Sec 3 and 4.4 ) conducted? What was the rubric, and how many experts were involved?

---

> ### Author Response · Authors · 2025-11-28
> **Rebuttal to Reviewer DLM4's comments (Part 1)**
>
> We want to thank the reviewer for conducting a thorough review. We appreciate the reviewer for recognizing the strengths of our paper and sharing valuable suggestions and questions.
>
> ## Addressing weaknesses
>
> ### Critical Lack of Reproducibility: This is the most significant flaw. The paper relies heavily on proprietary, unreleased models (e.g., "gpt-5", "o1", "o3"). Furthermore, it provides no details on the prompts, agent-to-agent interaction protocols, or stopping criteria, and makes no mention of code release. This makes the entire pipeline, which is the core contribution, unverifiable by the community.
>
> We appreciate the reviewer’s feedback. Following the reviewer’s suggestion, in the revised manuscript, we will commit two actions. First, we will release the entire framework as a public repository, which will include the full PyTorch implementation, the exact system prompts used to define the Theorist, Programmer, Critic, and Refiner roles, the detailed agent-to-agent communication protocols, and the precise stopping criteria for the refinement loop. Second, in the revised manuscript, we will add a new appendix section with a concrete, end-to-end worked example for the 1D Burgers' equation. This example will explicitly trace the pipeline’s operation: (1) the Theorist's output, showing its selection of an FNO backbone and its mathematical derivation for integrating an residual-Euler scheme to handle the viscous term for stability; (2) the Programmer's initial, self-contained PyTorch implementation of this FNO residual-Euler architecture ; and (3) the Critic's specific, actionable feedback, such as identifying a potential numerical instability in the discretization of the nonlinear advection term or a software issue like inefficient memory allocation in the spectral layer.
>
> ### Methodological Gaps in Theory-to-Architecture Mapping: The paper is vague on how the Theorist translates high-level mathematical principles into a concrete, novel neural architecture. This "theory-aware" step is the most innovative part of the pipeline, but it is treated as a black box. The paper lacks even a single, clear, worked example (e.g., for the 1D Burgers' equation) showing the chain from theory -> agent output -> architectural modification -> code.
>
> We thank the reviewer for the suggestion. We agree with the reviewer that the "theory-aware" mapping from mathematical principles to architectural code is our most critical innovation. To provide the essential clarity readers expect, as mentioned before, we will add a new appendix section with a concrete, end-to-end worked example for the 1D Burgers' equation. By doing this, we expect to show the processing chain and make the practical operation of our proposed method transparent.

---

### Author Response · Authors · 2025-12-03
**Summary of rebuttal discussions**

We thank the reviewers for their constructive feedback, which helped us clarify presentation, strengthen the discussions, and refine the characterization of our contributions and empirical scope.

## Key contributions

- We propose the first theory-aware, multi-agent LLM pipeline for fully automated neural operator design. The framework decomposes the complex design task into four cooperative roles that translate a PDE specification into a mathematically justified operator architecture, executable PyTorch implementation, adversarial correctness and stability checks, and targeted refinements.

- Rather than relying on unrestricted code generation, we constrain LLM reasoning through a theory-guided workflow: the LLM first selects a strong backbone from a curated operator “toolbox” and then proposes principled mathematical modifications tailored to the target PDE, which are subsequently stress-tested and corrected through Critic–Refiner loops.

- Across six benchmark PDE problems, the automatically synthesized operators consistently achieve accuracy matching or exceeding state-of-the-art human-designed baselines, while offering substantial parameter and training-time efficiency.

## Addressing comments

### Reproducibility

We will release the full framework as a public repository, including PyTorch code, all agent prompts, communication protocols, and stopping criteria. We will also add a complete worked example for the 1D Burgers’ equation, tracing the full pipeline from the Theorist’s mathematical formulation and backbone selection, to the Programmer’s initial code, and the Critic’s identification of numerical or implementation issues, followed by the Refiner’s corrections.

### Theory-to-architecture mapping

We clarified that the Theorist does not invent mathematics arbitrarily. A structured prompting strategy first provides a toolbox of existing neural operators, instructs backbone selection, and guides targeted theoretical adaptation based on PDE properties (stiffness, boundary conditions, conservation), with the Burgers’ example explicitly demonstrating the chain from theory to architecture to code.

### Explaining LLM effectiveness

We emphasized that success stems from constrained multi-agent reasoning, leveraging the LLM’s math-to-code inductive bias rather than free-form pattern matching. Ablations removing the Theorist increase errors by 3 to 5x, verifying the necessity of theory-aware design.

### Experimental scope and comparisons

We highlighted that evaluations cover both structured and irregular geometries (e.g., Elasticity and Airfoil) and compare against major human-designed operators. Direct comparison to automated PINN agents (e.g., PINNsAgent) was discussed as misaligned since PINNs solve single PDE instances rather than learn general operators, with follow-up testing planned when repositories become available.

### Design-time cost

We quantified design overhead: on 2D Navier–Stokes, full operator synthesis requires only 35 to 45 minutes on a single RTX 4090 over 7-9 agent iterations, versus days or weeks of expert manual development.

### Independent validation of theory quality

We clarified that the Theorist’s initial formulations were assessed by three independent PhD-level experts using binary correctness rubrics. Five of six tasks were judged theoretically correct before Critic intervention, with the challenging AFD case being the only exception.

### Robustness and generalization

We emphasized that the proposed framework has strong resolution generalization, robust performance on irregular geometries, and stability under noisy initial conditions across diverse PDE regimes.

---

### Meta-Review · Area_Chair_dpFe · 2026-01-07

**Summary:**

This paper proposes an automated framework using a multi-agent Large Language Model (LLM) pipeline to design neural operators for solving partial differential equations (PDEs). The authors introduce a four-agent pipeline consisting of a Theorist, Programmer, Critic, and Refiner. All three reviewers raised reasonable concerns and leaning towards rejection. The authors provided partial responses in the rebuttal phase, but some concerns remain unresolved. Therefore, I recommend rejection of this paper. I encourage the authors to revise the paper and resubmit it to future conferences.

**Reviewer Concerns:**

Reviewer xdJt raised concerns on the efficiency, incomplete baseline coverage, and missing experimental details. Authors provided some clarifications but did not address these concerns adequately with supporting experiments.

Reviewer zEVe questions the intuitions of using GPT-4 to design neural operators, and raised concerns on the missing experimental details and generalization ability of the designed models. Authors provided some clarifications but did not address these concerns adequately.

Reviewer DLM4 raised concerns on reproducibility, missing baselines, and insufficient result analysis. Authors provided some clarifications but did not address these concerns adequately.

**Reviewer Scores:**

Given that some concerns remain unresolved, I think the reviewers' scores remain the same.

---

### Decision · Program_Chairs · 2026-01-26

Reject